# Biomarker combination predicting imminent relapse after discontinuation of biological drugs in patients with rheumatoid arthritis in remission

**Eiji Sakashita[1], Katsuya Nagatani[2], Hitoshi Endo[1], Seiji Minota[2]***

1 Department of Biochemistry, Jichi Medical University School of Medicine, Tochigi, Japan, 2 Department of Medicine, Division of Rheumatology and Clinical Immunology, Jichi Medical University School of Medicine, Tochigi, Japan

☯ These authors contributed equally to this work.
* sminota@jichi.ac.jp

**Data Availability Statement:** All relevant data are within the paper and its Supporting Information files.

**Competing interests:** None

## Abstract

### Objectives

Compared to conventional disease-modifying antirheumatic drugs (DMARDs), biological DMARDs demonstrate superior efficacy but come with higher costs and increased infection risks. The ability to stop and resume biological DMARD treatment while maintaining remission would significantly alleviate these barriers and anxieties. The objective of this study was to identify biomarkers that can predict an imminent relapse, hopefully enabling the timely resumption of biological DMARDs before relapse occurs.

### Methods

Forty patients with rheumatoid arthritis who had been in remission for more than 12 months were included in the study. The patients discontinued their biological DMARD treatment and were monitored monthly for the next 24 months. Out of the 40 patients, 14 (35%) remained in remission at the end of the 24-month period, while 26 (65%) experienced relapses at different time points. Among the relapse cases, 13 patients experienced early relapse within 6 months, and another 13 patients had late relapse between 6 months and 24 months. Seventy-three cytokines in the sera collected longitudinally from the 13 patients with late relapse were measured by multiplex immunoassay. Using cytokines at two time points, immediately after withdrawal and just before relapse, volcano plot and area under the receiver operating characteristic curves (AUC) were drawn to select cytokines that distinguished imminent relapse. Univariate and multivariate logistic regression analyses were used for the imminent relapse prediction model.

### Results

IL-6, IL-29, MMP-3, and thymic stromal lymphopoietin (TSLP) were selected as potential biomarkers for imminent relapse prediction. All four cytokines were upregulated at imminent

relapse time point. Univariate and multivariate logistic regression showed that a combination model with IL-6, MMP-3, and TSLP yielded an AUC of 0.828 as top predictors of imminent relapse.

## Conclusions

This methodology allows for the prediction of imminent relapse while patients are in remission, potentially enabling the implementation of on- and off-treatments while maintaining remission. It also helps alleviate patient anxiety regarding the high cost and infection risks associated with biological DMARDs, which are the main obstacles to benefiting from their superb efficacy.

## Introduction

Patients with rheumatoid arthritis (RA) had suffered persistent pain from joint inflammation for a long period, culminating in joint deformity and miserable life until the advent of methotrexate. Methotrexate showed very good effects in reducing rheumatoid inflammation and pain and became an anchor drug in RA treatment [1–3]. However, a large number of patients did not enjoy its beneficial effect due to inadequate efficacy or an unacceptable level of side effects [4,5].

Biological disease-modifying antirheumatic drugs (bDMARDs) are made of monoclonal antibodies against inflammation-promoting cytokines. By blocking the actions of these cytokines physicochemically, bDMARDs mitigate joint inflammation in many patients with RA who are resistant to methotrexate. The number of patients who respond to and the degree of remission achieved by bDMARD are much greater than those of conventional DMARDs. If bDMARDs are introduced into treatment in the early stage of RA, many patients do not realise that they have RA. The sooner patients receive treatment, the better functional outcome ensues, as is true for any diseases [6]. However, there are several obstacles to starting bDMARDs in the early stage: cost and infection risk [7] from the patient's point of view.

In our previous article, we reported that it is possible to discriminate, with high probability at the time of bDMARD withdrawal, patients who will relapse from those who will remain in remission after bDMARD withdrawal by the relapse prediction index (RPI) calculated from the serum levels of IL-34, CCL1, IL-1β, IL-2 and IL-19 [8]. Our subsequent article showed that patients who relapse soon after bDMARD withdrawal (less than 6 months) are biochemically different from those who relapse long after withdrawal, and can be separated by measuring serum levels of INF-β at the time of bDMARD withdrawal [9].

In this article, we address whether biological markers can predict imminent relapses after bDMARD withdrawal before patients feel joint pain and swelling developing again. If this is possible, bDMARDs are restarted immediately in patients in need and on- and off-treatments with bDMARDs become possible while patients enjoy deep remission throughout.

## Materials and methods

### Study design and study population

This article used the same cohort of 40 RA patients previously reported who were recruited from February 4, 2010 through March 31, 2021 [8,9]. Briefly, forty patients with RA in long-term remission, i.e., for at least one year, had been followed monthly after bDMARD

withdrawal until exacerbation occurred or for 2 years if exacerbation did not occur. Serum samples were collected monthly, aliquoted, and stored at –80˚C until use from all 40 patients. Each patient's serum aliquot was thawed only once, and all samples were measured simultaneously to mitigate inter-measurement variation. Fourteen patients remained in remission, while twenty-six patients exacerbated at some time points. Among 26 patients who relapsed, 13 patients relapsed very early, i.e., within 6 months, after the bDMARD withdrawal, and another 13 relapsed late, i.e., after 6 months [9]. To find cytokines, if any, that could predict imminent relapse, cytokine levels were measured in sera collected monthly from 13 patients who relapsed late in the follow-up. In this paper, the term 'deep remission' is employed to characterize a level of remission that is akin to Boolean remission.

## Measurement of cytokines/chemokines

The Bio-Plex Pro human chemokine panel (40-plex, Bio–Rad Laboratories, Hercules, CA) and the Bio-Plex Pro human inflammation 1 panel (37-plex, Bio–Rad Laboratories) were used for the cytokine/chemokine measurements in the sera of patients as previously reported. There were several duplicates of cytokines in the two assay kits and finally 73 cytokines (S1 Table) were measured. Both assay kits contained heterophilic antibody blocking reagents to inhibit rheumatoid factor interference in the measurements.

## Protein–protein interaction analysis

Cytokines with significant differences between groups were used to elucidate the protein-protein interaction (PPI) network, and KEGG analyses using the STRING version 11.5 database (http://string-db.org) [10]. The PPI network was depicted using Cytoscape software version 3.9.1 (www.cytoscape.org) [11]. The STRING functional enrichment outputs on the KEGG pathway were created using GraphPad Prism 9.

## Ethics

This study was conducted in compliance with the Helsinki Declaration. The Jichi Medical University Institutional Review Board approved this study, and the patients gave their written informed consent before enrolling in the study. This study was registered in the University Hospital Medical Information Network Clinical Trials Registry (UMIN000044434).

## Statistical analysis

Statistical analyses were performed with EZR version 1.52 (Saitama Medical Centre, Jichi Medical University, Saitama, Japan) [12], which is a graphical user interface for R version 4.02 (The R Foundation for Statistical Computing, Vienna, Austria). Bio-Plex assay data were normalised by $\log_2$ transformation for volcano and violin plots. Analysis of the area under the ROC curve (AUC) and the logistic regression AUC over 10-fold cross-validation were performed using MetaboAnalyst 5.0 (www.metaboanalyst.ca) [13]. Kaplan–Meier survival curves were drawn and a log-rank test was performed to compare survival by imminent relapse prediction index (iRPI) score in the late relapse and non-relapse groups using GraphPad Prism 9 version 9.5.0 (www.graphpad.com).

## Results

### Patient demographics

We previously reported a cohort of 40 RA patients who were treated with bDMARD and had been in remission for at least one year [8,9]. bDMARDs were withdrawn from the patients and

**Table 1. Patients demographics.**

| Characteristics | Total population (n = 40) | | | |
|---|---|---|---|---|
| | Non-relapse (n = 14) | Early relapse (up to 6 months) (n = 13) | Late relapse (6 to 24 months) (n = 13) | p values |
| Age, years | 60 (39–63) | 59 (44–66) | 59 (45–66) | 0.976[a] |
| Female gender, n (%) | 10 (71.4) | 11 (84.6) | 11 (84.6) | 0.617[a] |
| Disease duration, years | 5.0 (3.0–7.5) | 5.0 (4.0–12.0) | 7.0 (6.0–11.0) | 0.224[a] |
| Radiographic stage III or IV[f], n (%) | 2 (14.3) | 6 (46.2) | 2 (15.4) | 0.106[a] |
| Number of bDMARDs used before study initiation, n (%) | | | | 0.068[a] |
| 1 | 13 (92.9) | 7 (53.8) | 10 (76.9) | |
| ≥2 | 1 (7.1) | 6 (46.2) | 3 (23.1) | |
| Remission duration before study initiation, months | 41.5 (25.8–52.0) | 44.0 (33.0–57.0) | 56.0 (24.0–62.0) | 0.567[a] |
| Methotrexate, n (%) | 11 (78.6) | 8 (61.5) | 11 (84.6) | 0.379[a] |
| dose, mg/week | 6.0 (4.0–8.0) | 4.0 (0.0–4.0) | 8.0 (6.0–8.0) | 0.093[a] |
| Prednisolone[g], n (%) | 0 (0.0) | 4 (30.8) | 0 (0.0) | 0.011[a, b] |
| dose, mg/day[g] | 0.0 (0.0–0.0) | 0.0 (0.0–1.0) | 0.0 (0.0–0.0) | 0.011[a, c] |
| Seropositive (RF or ACPA) before treatment with bDMARDs, n (%) | 13 (92.9) | 13 (100.0) | 10 (76.9) | 0.139[a] |
| CRP (mg/dL) | 0.05 (0.03–0.11) | 0.02 (0.02–0.03) | 0.03 (0.02–0.09) | NA[d] |
| SAA (μg/mL) | 0.0 (0.0–2.1) | 0.0 (0.0–0.0) | 0.0 (0.0–0.0) | NA[d] |
| DAS28-CRP before treatment with bDMARDs | 4.00 (3.50–4.38) | 4.25 (3.85–5.17) | 3.82 (2.68–4.06) | 0.450[a] |
| DAS28-CRP at study initiation | 1.29 (1.11–1.40) | 1.07 (1.04–1.12) | 1.13 (1.05–1.21) | 0.051[a] |
| Boolean remission, n (%) | 14 (100) | 13 (100) | 13 (100) | NA[e] |
| TNF inhibitors, n (%) | | | | 0.160[a] |
| infliximab | 4 (28.6) | 3 (23.1) | 6 (46.1) | |
| etanercept | 7 (50.0) | 6 (46.1) | 4 (30.8) | |
| adalimumab | 2 (14.3) | 0 (0.0) | 2 (15.4) | |
| IL-6 inhibitor, n (%) | | | | |
| tocilizumab | 1 (7.1) | 4 (30.8) | 1 (7.7) | |

Values are presented as medians (interquartile range) unless otherwise specified.

[a]Kruskal-Wallis test

[b]Non-relapse vs. early relapse: $p$ = 0.070; early relapse vs. late relapse: $p$ = 0.083 by Steel-Dwass' test

[c]Non-relapse vs. early relapse: $p$ = 0.071; early relapse vs. late relapse: $p$ = 0.085 by Steel-Dwass' test

[d]Not tested as all values were within the reference range

[e]Not tested as all values are equal

[f]Steinbrocker stage definition

[g]None of the patients in non-relapse group and late relapse group was taking prednisolone, and 4 in early relapse group were taking prednisolone at 1 mg in one, 2 mg in two and 3 mg in one at the study initiation. The amount of prednisolone was unchanged throughout the study period. Medians for methotrexate and prednisolone were drawn from all the patients on and off the medications. Statistical analysis was performed using EZR. SAA: Serum amyloid A. The upper limit of normal of SAA is 8.0 μg/mL. When the measurement value was < 8.0, 0 was assigned in this report.

they were followed monthly for 2 years. The patient demographics are shown in Table 1. Fourteen patients remained in remission for 2 years (non-relapse), 13 patients relapsed within 6 months (early relapse), and another 13 relapsed after 6 months (late relapse). There were no differences in baseline characteristics between the three groups, as shown in Table 1. Thirty-four and six patients were treated with TNF inhibitors and IL-6 inhibitor, respectively (Table 1). Only one patient was treated with the IL-6 inhibitor in the late relapse group, which was the target group analyzed in this report.

## Cytokine profile immediately before relapse in patients of late relapse

To identify a biomarker(s) indicative of imminent relapse in the late relapse group, we measured serum levels of cytokines in the sera of patients at the time of bDMARDs withdrawal ($L_{t1}$) and compared them with those immediately before relapse ($L_{t2}$) in the late relapse group (n = 13) (Fig 1). The quantification of 73 cytokines was performed with multiplex immunoassay systems. Data normalisation was carried out to reduce any systematic bias through the given data points and to provide a consistent biological comparison. Cytokines that were differentially regulated (fold change > 1.5) and statistically significant (t-test p value < 0.05) were filtered using the volcano plot (Fig 2A). We identified seven cytokines with a significant difference, thymic stromal lymphopoietin (TSLP), IL-29, IL-6, IL-27 p28, MMP-3, CCL17, and CXCL13, all of which were up-regulated at the $L_{t2}$ time point (Fig 2B).

In parallel, we performed the analysis of the area under the ROC curve (AUC) to evaluate the diagnostic performance of 73 cytokines for the prediction of imminent relapse by comparing the change in cytokine level at $L_{t1}$ and $L_{t2}$ in the late relapse group (Table 2). The highest AUC was found for IL-6 with an AUC of 0.831 (95% CI; 0.676–0.986). The other 5 cytokines, IL-29, TSLP, IFNγ, MMP-3, and TNFSF8, showed good AUC (AUC > 0.7).

To understand the protein–protein connections between the nine cytokines that were identified in the volcano plot or the AUC analysis (Fig 2A and Table 2), we obtained a potential protein–protein interaction (PPI) network using the STRING database and then analysed in Cytoscape (Fig 3A). The PPI network consisted of 14 nodes including 5 additional relevant proteins, and 47 edges. The proteins are represented as nodes, and the edges are interactions between the proteins. Most protein interactions were related to IL-6 (13 edges) and IFNγ (12 edges). To gain further insight into the signaling pathway that predicts imminent relapse, we further performed enrichment analysis using the Kyoto Encyclopedia of Genes and Genomes (KEGG) database. These up-regulated genes were mainly enriched in the IL-17 and JAK-STAT signaling pathways (Fig 3B).

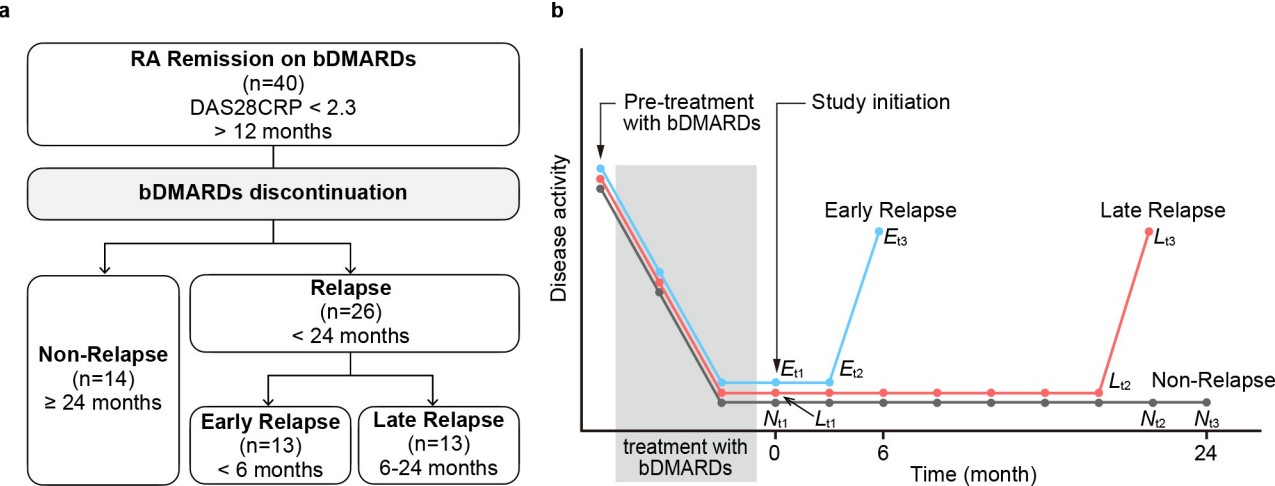

**Fig 1. Study design.** *a* Schematic presentation of the patients enroled and their outcome after bDMARD discontinuation. Forty RA patients in remission judged by DAS28-CRP for more than 12 months by using bDMARDs were included. After discontinuation of bDMARD, 14 patients remained in remission for at least two years (Non-Relapse) and 26 relapsed at some time point (Relapse). Relapsed patients were dichotomised into those who relapsed less than 6 months (Early Relapse, n = 13) and those who relapsed after 6 months (Late Relapse, n = 13). *b* Timing of clinic visits and serum sampling. The cyan line indicates late relapse group. The orange-red line indicates late relapse group. The grey line indicates non-relapse group. $E_{t1}$, $L_{t1}$ and $N_{t1}$: Time of study initiation in early relapse (E), late relapse (L) and non-relapse (N); $E_{t2}$, $L_{t2}$ and $N_{t2}$: Time of the last confirmed remission in relapse groups or just before the study end for non-relapse group; $E_{t3}$, $L_{t3}$, and $N_{t3}$: Time of first confirmed relapse in relapse groups and study end for non-relapse group. The images were created using Adobe Illustrator (ver. 27.0.1, https://www.adobe.com/).

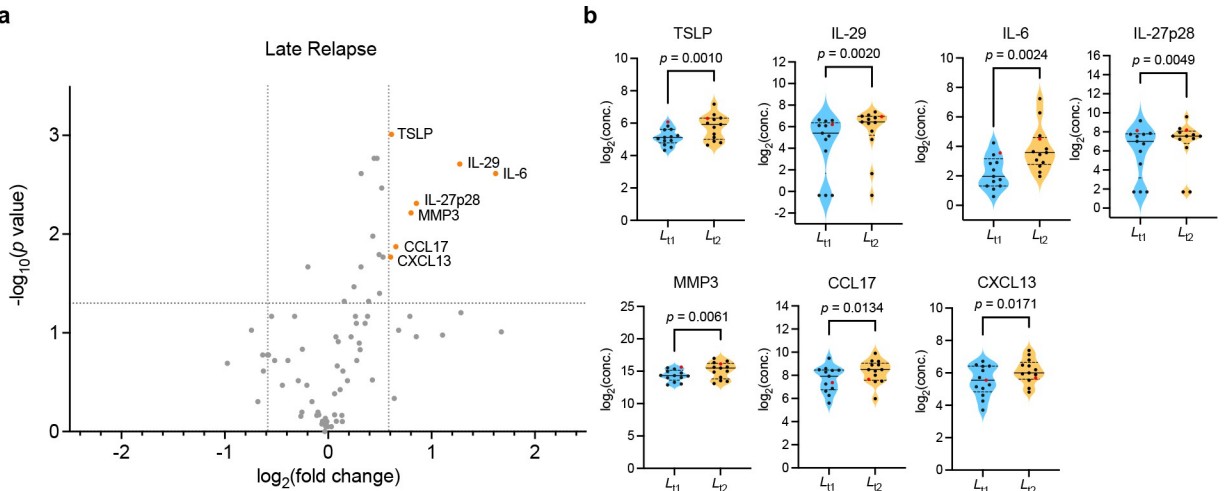

**Fig 2. Volcano plot showing cytokines with a significant *p* value (y-axis) and a fold change ($L_{t2}/L_{t1}$, x-axis) between sampling time points at $L_{t1}$ and $L_{t2}$ in late relapse group.** *a* Volcano plot of late relapse group. Cytokines with a $\geq$ 1.5-fold change (vertical dotted lines) and a *p* value of < 0.05 (horizontal dotted line) are shown by orange-red dots. *b* Violin plot showing the distribution of $\log_2$-transformed cytokines selected by volcano plot at time points $L_{t2}$ (orange) and $L_{t1}$ (light blue) in late relapse group. Black and red dots indicate patients treated with TNF and IL-6 inhibitors, respectively. The non-parametric Wilcoxon test was used for the group comparison. The images were created using GraphPad Prism 9 (www.graphpad.com).

## Imminent relapse prediction model

We tested the performance of all possible combinations of 4 cytokines, *i.e.* IL-6, TSLP, IL-29, and MMP-3, which were identified in both the volcano and AUC analyses, using logistic regression AUC over 10-fold cross-validation (15 combinations in total, Table 3). The three-cytokine combination model consisting of IL-6, TSLP, and MMP-3 showed the best performance among all combinations tested (AUC = 0.828, Fig 4A). The predicted probability of imminent relapse using the three-cytokine model (termed "imminent relapse prediction index score; iRPI score") can be calculated as follows: iRPI score = logit (*P*) = 10.576 + 1.018 (*IL-6*) + 6.344 (*TSLP*) – 3.279 (*MMP-3*), where *IL-6*, *TSLP* and *MMP-3* were serum concentrations transformed in $\log_2$. The formula for the iRPI score, *i.e.* logit *(P)*, was established via a logistic regression model, where *P* is the estimated probability of imminent relapse. The cut-off value was 0.410. That is, if the iRPI score at some time point becomes $\geq$ 0.410, the patient is predicted to relapse very soon thereafter.

To evaluate whether iRPI is an effective model to predict imminent relapse in patients who maintained remission for more than 6 months after bDMARD withdrawal, iRPI was compared at two time points, t1 and t2, in the two groups, *i.e.*, non-relapse ($N_{t1}$ and $N_{t2}$) and late relapse ($L_{t1}$ and $L_{t2}$). In the late relapse group, the iRPI score was significantly higher at $L_{t2}$

**Table 2. Top 6 cytokines in AUC analysis.**

| cytokine | AUC | 95% CI | *p* value |
|---|---|---|---|
| IL-6 | 0.831 | 0.676–0.986 | 0.004 |
| IL-29 | 0.734 | 0.535–0.932 | 0.043 |
| TSLP | 0.731 | 0.530–0.932 | 0.046 |
| IFNγ | 0.713 | 0.513–0.913 | 0.065 |
| MMP-3 | 0.710 | 0.505–0.916 | 0.069 |
| TNFRSF8 | 0.704 | 0.496–0.912 | 0.077 |

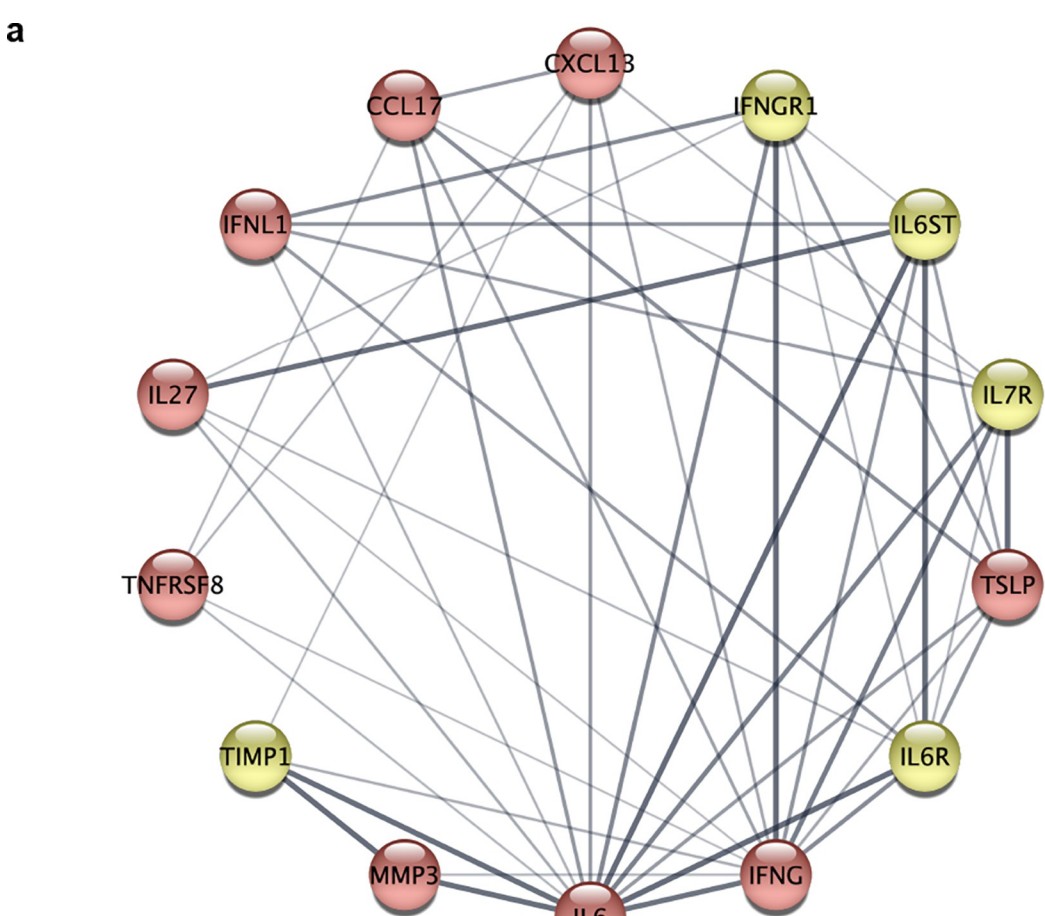

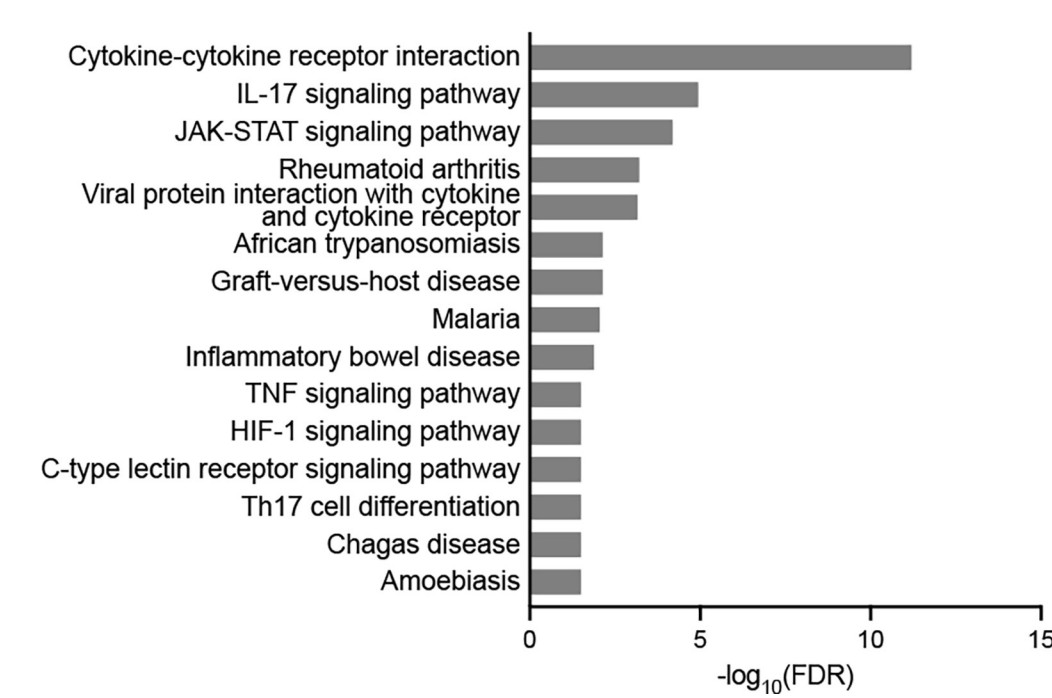

**Fig 3. Biomolecular network associated with late relapse.** *a* Protein–protein interaction (PPI) networks of 9 target proteins associated with late relapse shown in red were analyzed by STRING database with a confidence score of 0.4, and five more targets were identified and shown in yellow. Line thickness indicates the strength of data support. *b* Top 15 pathways from KEGG pathway analysis of 9 proteins associated with late relapse. X-axis indicates the significance ($-\log_{10}$[FDR]) of the pathway association, in bars. The PPI image was created using Cytoscape (ver. 3.9.1, https://cytoscape.org). The image of pathway analysis was created using GraphPad Prism 9 (www.graphpad.com).

than at $L_{t1}$, with $p = 0.0002$ by the non-parametric Wilcoxon test (Fig 4B). In the non-relapse group, the difference between $N_{t1}$ and $N_{t2}$, was not significant. Note that the iRPI score at $L_{t2}$ was higher than that at $L_{t1}$ in each patient with late relapse (Fig 4C). All the relevant data from each patient used in this study are provided as S2 Table and S2 Fig.

Next, it was examined whether iRPI has the ability to select patients with imminent relapse among the patients in the combined group of late relapse (n = 13) and non-relapse (n = 14) using Kaplan–Meier analysis with the log-rank test. Kaplan–Meier analysis showed a significant difference in the sustained remission rate between patients with iRPI ≥ cut-off (n = 13) and those with iRPI < cut-off (n = 14) (log-rank test, $p = 0.0020$) (Fig 4D). This suggests that the iRPI model would have the power to predict imminent relapse, or relapse *per se*, among patients who remain in remission beyond 6 months.

**Table 3. Logistic regression AUC over 10-fold cross-validation on each feature and their combinations.**

| Cytokine(s) | AUC (95% CI) | Sensitivity (95% CI) | Specificity (95% CI) |
|---|---|---|---|
| IL-6, TSLP, MMP3 | 0.828 (0.665–0.992) | 0.846 (0.846–1.000) | 0.769 (0.540–0.998) |
| IL-6, TSLP, IL-29 | 0.822 (0.658–0.987) | 0.692 (0.692–0.943) | 0.923 (0.778–1.000) |
| IL-6 | 0.814 (0.647–0.980) | 0.769 (0.769–0.998) | 0.769 (0.540–0.998) |
| IL-6, IL-29 | 0.811 (0.646–0.975) | 0.692 (0.692–0.943) | 0.769 (0.540–0.998) |
| IL-6, IL-29, MMP3 | 0.811 (0.643–0.978) | 0.692 (0.692–0.943) | 0.769 (0.540–0.998) |
| IL-6, MMP3 | 0.799 (0.621–0.977) | 0.846 (0.846–1.000) | 0.769 (0.540–0.998) |
| IL-6, TSLP, IL-29, MMP3 | 0.799 (0.620–0.978) | 0.769 (0.769–0.998) | 0.769 (0.540–0.998) |
| IL-6, TSLP | 0.799 (0.616–0.981) | 0.846 (0.846–1.000) | 0.769 (0.540–0.998) |
| TSLP, MMP3 | 0.751 (0.553–0.950) | 0.692 (0.692–0.943) | 0.769 (0.540–0.998) |
| TSLP, IL-29, MMP3 | 0.746 (0.546–0.945) | 0.769 (0.769–0.998) | 0.769 (0.540–0.998) |
| TSLP | 0.683 (0.459–0.907) | 0.615 (0.615–0.880) | 0.846 (0.650–1.000) |
| IL-29, MMP3 | 0.675 (0.458–0.891) | 0.615 (0.615–0.880) | 0.769 (0.540–0.998) |
| TSLP, IL-29 | 0.675 (0.458–0.891) | 0.615 (0.615–0.880) | 0.769 (0.540–0.998) |
| IL-29 | 0.675 (0.455–0.894) | 0.615 (0.615–0.880) | 0.846 (0.650–1.000) |
| MMP3 | 0.669 (0.447–0.890) | 0.692 (0.692–0.943) | 0.692 (0.441–0.943) |

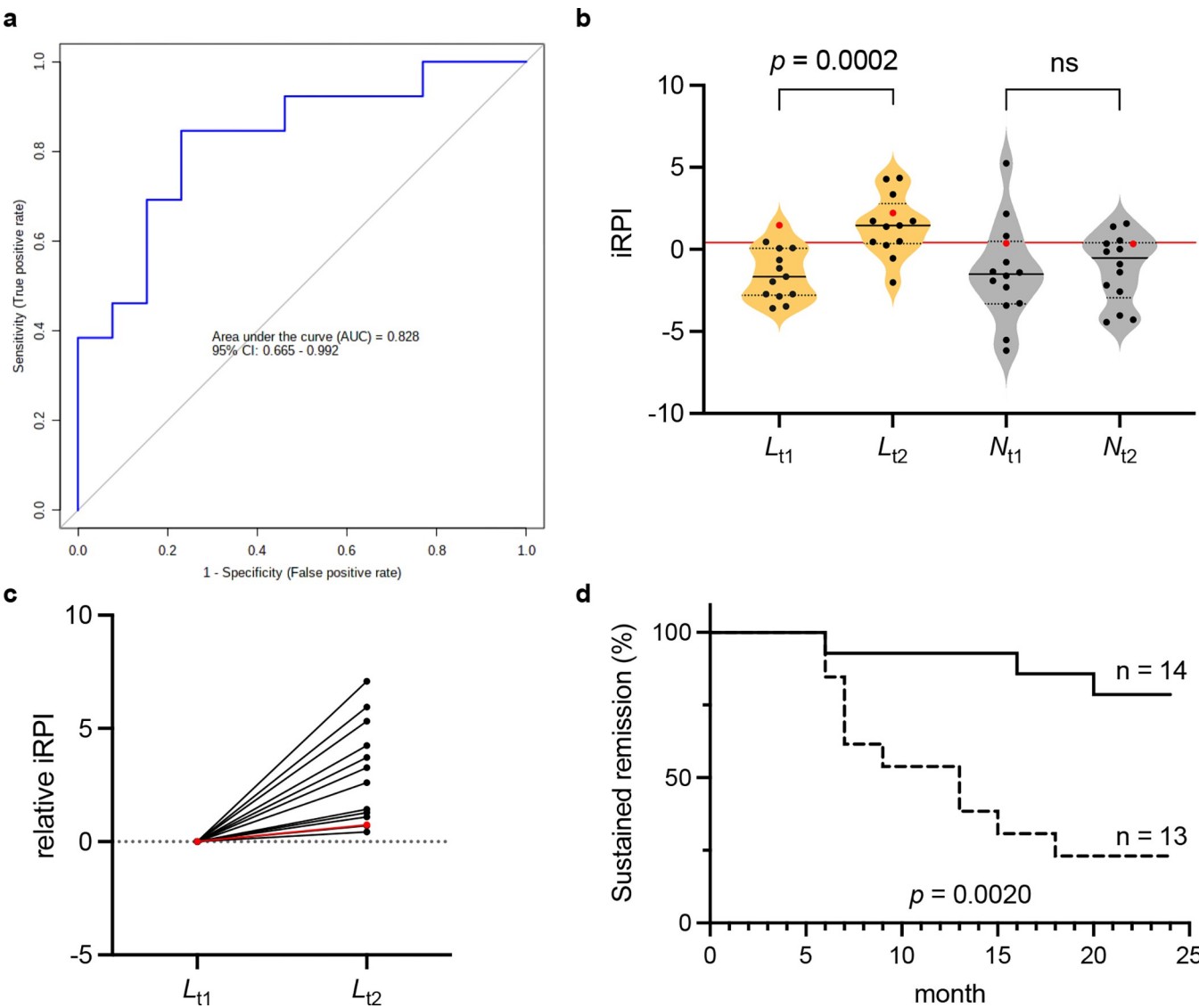

**Fig 4. Performance test of three-cytokine combination model for prediction of imminent relapse after bDMARDs withdrawal.** *a* ROC curve for logistic regression using IL-6, TSLP, and MMP-3 combination model. The area under the curve (AUC) for the ROC curves is annotated with the 95% confidence interval (CI) by the Wilson/Brown method. *b* Violin plot showing the distribution of iRPI score at time points $L_{t1}$ and $L_{t2}$ in late relapse group (orange), and $N_{t1}$ and $N_{t2}$ in non-relapse group (grey). Black and red dots indicate patients treated with TNF and IL-6 inhibitors, respectively. The red line indicates iRPI cut-off value (0.41). ns: Not significant. *c* Change in iRPI at $L_{t1}$ and $L_{t2}$ in patients in late relapse group. The differential was shown here in each patient by subtracting iRPI at $L_{t1}$ from iRPI at $L_{t2}$. Black dot/line and red dot/line indicate patients treated with TNF and IL-6 inhibitors, respectively. *d* Kaplan–Meier curves of sustained remission for 24 months, stratified by the iRPI cut-off value at either $L_{t2}$ or $N_{t2}$. Fourteen patients were low in iRPI at $N_{t2}$ (solid line) and 13 patients were high in iRPI at $L_{t2}$ (dotted line). The images were created using GraphPad Prism 9 (www.graphpad.com).

## Discussion

The great efficacy of bDMARDs in the field of rheumatology has provided RA patients with a much better quality of life and a brighter future, which invigorated not only patients but also rheumatologists [14,15]. Before the advent of bDMARDs, the rheumatology clinic was gloomy; It was seldom possible, if ever, to find effective drugs except methotrexate [1]. Today, rheumatologists recommend, actively and confidently, RA patients to take one of bDMARDs. However, there are several obstacles to overcome before reluctant patients willingly accept bDMARDs; the cost and infection risk are much higher than those of conventional DMARDs.

When starting bDMARDs, one of the most frequently asked questions in the rheumatology clinic is whether patients need to use them for a long period, if not forever, or whether they can discontinue them after reaching remission. Because we did not have an answer to this question, the practical way for us to take is to discontinue bDMARD, wait and see, and rush to resume bDMARD when RA flares up. Approximately 65% of the patients had relapses by two years after bDMARD withdrawal even if they reached remission of the Boolean criteria level [8,16]. Patients suffer joint pain and swelling, and cartilage and bone destruction progresses further once relapse occurs. In our previous article, five cytokines, IL-34, CCL1, IL-1β, IL-2 and IL-19, can distinguish, at the time of bDMARD withdrawal with good probability, patients who relapse within 2 years from those who remain in remission [8].

In a subsequent article, we reported that there are two types of relapse pattern: early (within 6 months after bDMARDs withdrawal) and late (after 6 months) relapse [9]. Approximately half of the relapses occurred within 6 months, and the slope of the Kaplan–Meier curve was very steep compared to that of late relapses. The serum level of only IFNβ, at the time of bDMARDs withdrawal, was higher in patients who relapsed early compared to those who relapsed late. A higher serum IFNβ level was maintained in the early relapse group throughout the observation period until the actual relapse occurred; IFNβ could be a good discriminator. Given the brief interval between bDMARD withdrawal and the onset of actual relapse, we advise against discontinuing bDMARDs for patients in the early relapse group, even if they are in complete remission. However, this recommendation requires prospective verification. Patients who are classified as relapse late (*i.e.*, at the time of bDMARD withdrawal, the RPI predicts relapse, but the IFNβ level is low) have a chance to stop bDMARD at least for some time. However, the resumption of bDMARD is too late once patients have symptoms of flare-ups.

In this report, we address whether imminent relapse can be predicted before patients and rheumatologists even notice it. Forty patients who entered remission with bDMARD for a long period (more than 1 year) were closely followed at 1-month intervals, after bDMARD withdrawal with their serum samples stored until the actual relapse occurred. By screening 73 biomarkers in each serum sample, retrospectively, IL-6, MMP-3, and thymic stromal lympho-poietin (TSLP) were selected as prediction markers of imminent relapses. If the logit (*P*) calculated from these three biomarkers became $\geq 0.41$, they had relapses very soon afterwards (from 1 to several follow-up intervals). We recommend the resumption of bDMARD to patients at this moment even if they have no symptoms to prevent relapses. However, prospective verification is necessary.

IL-6 is directly involved in joint inflammation and IL-6 receptor inhibitors are used as therapeutics. When IL-6 receptor inhibitors are infused into RA patients with active disease, their serum levels of IL-6 increase greatly and rapidly due to the continued production of IL-6 by active inflammation and the capping of the IL-6 receptor by IL-6 inhibitor [17]. In our relevant patients, there was one patient each in the late relapse and non-relapse group who was treated with the IL-6 inhibitor. Serum levels of IL-6 in both patients did not differ significantly from those treated with TNF inhibitor when they reached deep remission; IL-6 production decreased to the level that receptor capping by the IL-6 inhibitor did not influence the serum level of IL-6 much (S1 Fig). Therefore, it makes sense to apply iRPI that includes IL-6 to patients treated with IL-6 inhibitor as long as they maintain deep remission (S1 Data).

MMP-3 is a proteinase produced by synovial cells and chondrocytes and degrades the extra-cellular matrix such as proteoglycan, fibronectin, and collagen [18]. Increased production of MMP-3 contributes to cartilage destruction [19]. The serum level of MMP-3 is sometimes used as a marker of RA disease activity [20]. Therefore, it is quite reasonable for serum levels of MMP-3 to increase before relapse occurs.

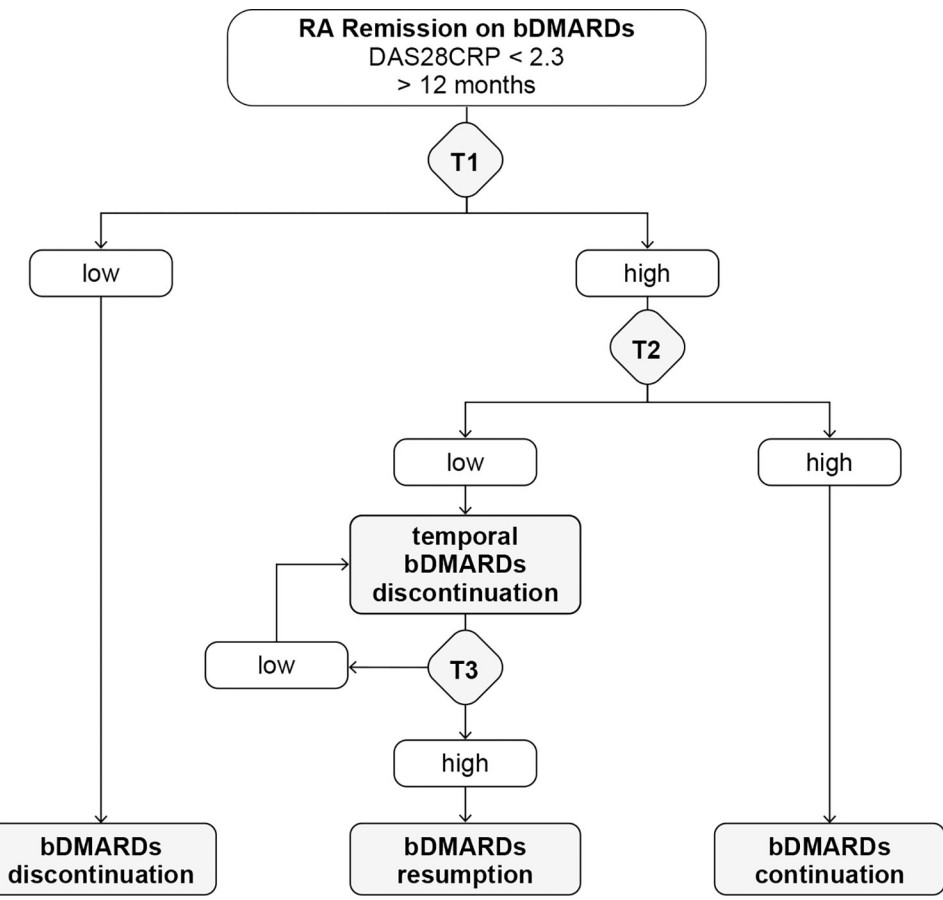

**Fig 5. Algorithm of bDMARD withdrawal in RA patients in long-term remission.** When bDMDRD withdrawal is considered, calculate the relapse prediction index (RPI) at T1. If RPI is low, bDMARDs can be withdrawn. If RPI is high, measure IFNβ at T2. If IFNβ is high, continue bDMARDs. If IFNβ is low, temporal discontinuation of bDMARDs is possible and measure the imminent relapse prediction index (iRPI) at T3. Repeat this process and resume bDMARDs when the iRPI becomes high. The image was created using Adobe Illustrator (ver. 27.0.1, https://www.adobe.com/).

TSLP and IL-7 have, in common, IL-7Rα as one of the receptor subunits. The other subunit of the IL-7 receptor is the common γ chain and the TSLP counterpart is the TSLP receptor. IL-7 is one of the members of the IL-2 superfamily, and after binding to the receptor, IL-7 acts as a proliferation and differentiation factor for T cells, B cells and innate lymphoid cells by activating JAKs and STATs [21,22]. It might also work as a tumor promoting or suppressing factor depending on the tumors [23]. On the other hand, several lines of evidence suggest that TSLP plays a pivotal role in allergic diseases [24,25]. Th2 cells and cytokines secreted by them, such as IL-4, IL-5, and IL-13 are the main players in allergic reactions. In addition to allergy, TSLP is found to be abundant in autoimmune diseases such as psoriasis and rheumatoid arthritis [26]. RA patients have higher levels of TSLP in synovial fluids than osteoarthritis patients, and the TSLP receptor is overexpressed in myeloid dendritic cells of synovial fluids in RA patients [27,28]. Although the role of TSLP in RA inflammation has not been elucidated, higher levels of serum TSLP in patients with imminent relapse could be important not only as a predictor of relapse, but also as a factor related to RA pathogenesis; pathological process from deep remission to relapse could be reminiscent of the process from healthy status to the onset of RA. It might be noted that the pathway analysis using KEGG showed that the late relapse was

strongly related to IL-17 signaling and JAK-STAT signaling pathways, which are important for autoimmune inflammation, as well as an innate immune response such as trypanosomiasis or malaria [29,30].

Finally, we propose the algorithm when bDMARD withdrawal is considered (Fig 5). The deeper the remission, the greater the chance of safe bDMARD withdrawal [31]. Therefore, it is preferable for patients to keep remission longer, *e.g.*, more than 1 year. Measure serum levels of IL-34, CCL1, IL-1β, IL-2, and IL-19 when bDMARD withdrawal is considered and calculate the relapse prediction index (RPI) from these five cytokines [8]. If the index is < 0.63, the probability of future relapse seems to be minimal (T1 in Fig 5). If the RPI is ≥ 0.63, then measure the serum level of IFNβ (T2 in Fig 5). If the IFNβ level is ≥ 3.38 in $\log_2$, relapse is highly probable less than 6 months after bDMARD withdrawal and it is recommended that patients continue with bDMARD. If the level of IFNβ is < 3.38 in $\log_2$, it is predicted that relapse, if any, occurs after 6 months of bDMARD withdrawal. In these cases, there is a chance to withdraw the bDMARD, at least temporarily. Follow the patients closely and regularly, preferably at 1-month intervals, and measure serum levels of IL-6, TSLP, and MMP-3 and calculate the iRPI therefrom. If the patients are in clinical remission and the iRPI is < 0.41, follow and repeat the measurements (T3 in Fig 5). If the iRPI becomes ≥ 0.41, resume bDMARD, even if the patients are in clinical remission. It is highly probable that relapses are imminent. However, the primary limitation of this study is the relatively small sample size of patients studied, and further validation is essential to confirm the results.

The approach we presented here when bDMARD withdrawal is considered in RA patients in remission is novel and needs to be consolidated by a large population of patient. This methodology might be applicable not only to bDMARDs but also to small molecular antirheumatic drugs such as JAK inhibitors of high cost and infection risk along with soft association with venous thromboembolism in some patient population [32]. Beyond rheumatology, it will also be helpful in other medical disciplines where biological drugs are used in common to implement on- and off-treatments and deprescription to reduce patients' costs and concerns.

## Supporting information

**S1 Fig. Serum IL-6 concentration in patients of late relapse and non-relapse groups treated with either IL-6 inhibitor (IL-6i) or TNF inhibitor (TNFi).** $\log_2$-transformed IL-6 concentrations in patients immediately after bDMARD withdrawal. Orange-red and grey dots indicate patients of late relapse and non-relapse, respectively. Nonparametric Mann-Whitney test was used for the group comparison. Significance is defined by $p < 0.05$. ns, not significant. (TIF)

**S2 Fig. The trend of DAS28-CRP from the time point of biologics discontinuation to relapse in the 13 cases of late relapse.** Paired t-test was employed for the group comparison, and the p-values were adjusted using the Bonferroni method. (TIF)

**S1 Table. Inflammatory and chemokine biomarkers measured.** (PDF)

**S2 Table. DAS28-CRP and its components from the time point of biologics discontinuation to relapse in the 13 cases of late relapse.** (XLSX)

**S1 Data. Summary of Bio-Plex analyte concentration (pg/mL).** (XLSX)

## Acknowledgments

The authors would like to thank all the patients who participated in this study. The authors also thank Dr. Takamasa Murosaki, Dr. Natsuki Shima, and Dr. Hiroi Kusaka for patient care, and Ms. Chiyomi Hayashi, Ms. Sachiko Mamada, and Ms. Chisato Udagawa for excellent technical assistance. They are all the staff of Jichi Medical University.

## Author Contributions

**Conceptualization:** Seiji Minota.

**Data curation:** Eiji Sakashita, Katsuya Nagatani, Hitoshi Endo.

**Formal analysis:** Eiji Sakashita, Hitoshi Endo.

**Funding acquisition:** Seiji Minota.

**Investigation:** Eiji Sakashita, Katsuya Nagatani, Seiji Minota.

**Methodology:** Eiji Sakashita, Katsuya Nagatani, Hitoshi Endo, Seiji Minota.

**Project administration:** Seiji Minota.

**Resources:** Katsuya Nagatani.

**Supervision:** Hitoshi Endo, Seiji Minota.

**Validation:** Eiji Sakashita.

**Visualization:** Eiji Sakashita.

**Writing – original draft:** Eiji Sakashita, Seiji Minota.

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
