## [Decision Letter · Decision Letter 0]

18 Dec 2023

PONE-D-23-22054Biomarker combination predicting imminent relapse after discontinuation of biological drugs in patients with rheumatoid arthritis in remissionPLOS ONE

Dear Dr. Minota,

Thank you for submitting your manuscript to PLOS ONE. After careful consideration, we feel that it has merit but does not fully meet PLOS ONE’s publication criteria as it currently stands. Therefore, we invite you to submit a revised version of the manuscript that addresses the points raised during the review process. Specifically, both reviewers found some interest in this study, but pointed ut a number of serious issues to be improved or amended. I ask the authors to fully respond to all comments made by reviewers in the revised version.  Please ensure that your decision is justified on PLOS ONE’s publication criteria and not, for example, on novelty or perceived impact.

We look forward to receiving your revised manuscript.

Kind regards,

Masataka Kuwana, MD, PhD

Academic Editor

PLOS ONE

Journal Requirements:

"No"

Reviewers' comments:

Reviewer's Responses to Questions

**Comments to the Author**

1. Is the manuscript technically sound, and do the data support the conclusions?

Reviewer #1: Yes

Reviewer #2: No

2. Has the statistical analysis been performed appropriately and rigorously? 

Reviewer #1: Yes

Reviewer #2: Yes

3. Have the authors made all data underlying the findings in their manuscript fully available?

Reviewer #1: Yes

Reviewer #2: Yes

4. Is the manuscript presented in an intelligible fashion and written in standard English?

Reviewer #1: Yes

Reviewer #2: Yes

5. Review Comments to the Author

Reviewer #1: The authors described biomarkers and algorithm to detect relapse just prior to the onset of clinical symptoms 6 months or later after discontinuation of biologics in patients with RA in remission in a previously reported population. This manuscript is interesting and addresses a clinical unmet need. From the reviewer’s perspective, I have a few comments that would be worth addressing.

-I suggest that the authors show the definition of the terms, “deep remission”, “long-term remission” and “relapse very early”, clearly in the section of Material and Methos.

-Considering the period of the study indicated, the cryopreservation period of serum is likely to be around 5 to 10 years. It is suggested that a clear indication of how the stability of stored serum cytokines in ensured.

- As the results would be more convincing if clinical parameters at the time of imminent relapse were presented, I suggest that the authors show the trends of disease activity from the time of biologics discontinuation to relapse in the 13 cases, and in particular disease activity and its components at Lt2. Please also indicate the time between Lt2 and clinical relapse.

- The authors recommend re-administering biologics when the biomarker score exceeds the cut-off, even if there are no symptoms, but the results do not show whether re-administering biologics really reduces relapse, so I suggest that this should be stated as a limitation.

- In addition, the fact that the results of this study were derived from a limited population of 13 cases only and that no validation cohort was set up is suggested to be stated as a limitation of this study.

- In Discussion section, p.21, line 4, the sentences “The deeper the remission ~ more than 1 year.” should be added with supporting citations.

- P.12, line 3, “IL-19” is considered a typographical error for IL-29.

Reviewer #2: This study aimed to determine whether biomarkers could predict relapse after withdrawal of biological DMARDs (bDMARDs) in patients with rheumatoid arthritis (RA) who had been in remission for more than 12 months on bDMARDs. The imminent relapse prediction index (iRPI) > 0.4 with AUC 0.828 was found to predict imminent relapse.The data is valuable because blood samples were collected monthly from 40 RA patients who had withdrawn from bDMARDs. However, this study has several problems:

1. The abstract states that "The objective of this study was to identify biomarkers that can predict an imminent relapse, enabling the timely The objective of this study was to identify biomarkers that can predict an imminent relapse, enabling the timely resumption of biological DMARDs before relapse occurs.” This study has developed an index to predict imminent recurrence, but it is not known whether resumption of bDMARDs will prevent RA recurrence. If the iRPI is high at T3 in Figure 5 of the Discussion, the authors suggest resuming bDMARDs, but the validity of resuming bDMARDs has not been investigated.

2. The information about Patients who maintained DAS28-CRP<2.3 for more than 12 months but did not withdraw bDMARDs were not provided in this study. The physician's decision to withdraw bDMARDs may have led to selection bias. I reviewed the methods of previous reports but could not find out if this study involved a consecutive patients. (were there any patients who discontinued abatacept?). Please provide the number of patients who have maintained DAS28-CRP score of less than 2.3 for over 12 months and the number of those patients who have discontinued bDMARDs.

3. Is the change in biomarkers after withdrawal of TNF inhibitors and IL-6 inhibitors the same? In the discussion section, the authors argue that it is reasonable to include patients who were treated with IL-6 inhibitors in this study because there is no difference in IL-6 levels at the time of withdrawal of bDMARDs. "However, it is not guaranteed that biomarker changes observed following the discontinuation of TNF inhibitors and IL-6 inhibitors will exhibit similar patterns. I suggest that TNF inhibitors and IL-6 inhibitors should be analysed separately.

6. PLOS authors have the option to publish the peer review history of their article (what does this mean?). If published, this will include your full peer review and any attached files.

Reviewer #1: No

Reviewer #2: **Yes: **Shinji Watanabe

---

## [Author Response · Author response to Decision Letter 0]

26 Jan 2024

Dear Reviewers,

We sincerely appreciate your thorough review of our manuscript and the valuable feedback you provided, which significantly contributed to enhancing the quality of our paper. We have meticulously revised the document, carefully incorporating each of your insightful suggestions and addressing all your queries in detail.

Our replies to each individual comment are highlighted in red.

REVIEWER #1

1. I suggest that the authors show the definition of the terms, “deep remission”, “long-term remission” and “relapse very early”, clearly in the section of Material and Methos.

As you recommended, the definitions were included in the Material and methods. (L93~L94 for “deep remission”, L84 for “long-term remission”, L90 for “relapse very early”)

2. Considering the period of the study indicated, the cryopreservation period of serum is likely to be around 5 to 10 years. It is suggested that a clear indication of how the stability of stored serum cytokines in ensured.

We added the following sentence in the Materials and Methods section. 

Each patient's serum aliquot was thawed only once, and all samples were measured simultaneously to mitigate inter-measurement variation. (L87~L88)

3. As the results would be more convincing if clinical parameters at the time of imminent relapse were presented, I suggest that the authors show the trends of disease activity from the time of biologics discontinuation to relapse in the 13 cases, and in particular disease activity and its components at Lt2. Please also indicate the time between Lt2 and clinical relapse.

As per your request, we are pleased to present a table (referred to as S2 Table) containing clinical parameters comprising DAS28-CRP measurements at Lt1, Lt2, and Lt3, as well as the time intervals from Lt2 to Lt3 in days for each of the 13 patients. While the S2 Fig can be derived from the parameters listed in the S2 Table, we have included it for your convenience. We hope that the table and figure will assist you in comprehending the clinical trends you were interested in. (L252~253)

4. The authors recommend re-administering biologics when the biomarker score exceeds the cut-off, even if there are no symptoms, but the results do not show whether re-administering biologics really reduces relapse, so I suggest that this should be stated as a limitation.

Your comment is quite appropriate. We addressed several limitations in the Discussion section. Please see L288~291, L304, and L355~356.

5. In addition, the fact that the results of this study were derived from a limited population of 13 cases only and that no validation cohort was set up is suggested to be stated as a limitation of this study.

Your comment is accurate. We have recognized and emphasized the constraints of the limited sample size and the absence of validation in our study. (L355~356)

6. In Discussion section, p.21, line 4, the sentences “The deeper the remission ~ more than 1 year.” should be added with supporting citations.

We added a new reference (#31) as you recommended. (L341)

7. P.12, line 3, “IL-19” is considered a typographical error for IL-29.

We corrected the error. Thank you for pointing this out. (L160, L191)

REVIEWER #2

1. The abstract states that "The objective of this study was to identify biomarkers that can predict an imminent relapse, enabling the timely The objective of this study was to identify biomarkers that can predict an imminent relapse, enabling the timely resumption of biological DMARDs before relapse occurs.” This study has developed an index to predict imminent recurrence, but it is not known whether resumption of bDMARDs will prevent RA recurrence. If the iRPI is high at T3 in Figure 5 of the Discussion, the authors suggest resuming bDMARDs, but the validity of resuming bDMARDs has not been investigated.

Thank you for your valuable feedback. We have incorporated the limitations you mentioned into three sections of the manuscript. (L23) (L288~291) (L304)

2. The information about Patients who maintained DAS28-CRP<2.3 for more than 12 months but did not withdraw bDMARDs were not provided in this study. The physician's decision to withdraw bDMARDs may have led to selection bias. I reviewed the methods of previous reports but could not find out if this study involved a consecutive patients. (were there any patients who discontinued abatacept?). Please provide the number of patients who have maintained DAS28-CRP score of less than 2.3 for over 12 months and the number of those patients who have discontinued bDMARDs.

Only patients who achieved and maintained remission (DAS28-CRP < 2.3) for a minimum of one year were deemed eligible for inclusion in the study. From the pool of eligible candidates, we ascertained their preference for discontinuing bDMARDs. For those expressing a desire to discontinue, we provided a comprehensive explanation of the study design and obtained informed consent from those willing to participate. The number of patients who opted to either continue bDMARD treatment or discontinue without participating in the study was not recorded or retrievable. It is important to note that patients treated with abatacept did not meet the inclusion criteria for this study and, therefore, were not included.

3. Is the change in biomarkers after withdrawal of TNF inhibitors and IL-6 inhibitors the same? In the discussion section, the authors argue that it is reasonable to include patients who were treated with IL-6 inhibitors in this study because there is no difference in IL-6 levels at the time of withdrawal of bDMARDs. "However, it is not guaranteed that biomarker changes observed following the discontinuation of TNF inhibitors and IL-6 inhibitors will exhibit similar patterns. I suggest that TNF inhibitors and IL-6 inhibitors should be analysed separately.

In the present study, we analyzed 13 patients in the late-relapse group along with 14 patients in the non-relapse group to identify characteristic cytokine markers indicative of imminent relapse. However, we have included pertinent cytokine data from all 40 patients at time points t1 and t2 in the form of a supplementary data sheet (S1 Data). Patients treated with the IL-6 inhibitor comprised R26 in the late-relapse group, N10 in the non-relapse group, as well as R7, R17, and R20 in the early relapsed group (We added “inhibitor” column in S1 Data). There were no discernible trends in the longitudinal cytokine kinetics among users of TNF inhibitors and IL-6 inhibitors. We would like to include the data from IL-6 inhibitor users in the induction of iRPI. See supplementary data sheet S1 Data in L315.

The focus of this study is not on the disease process that improves with the use of bDMARDs but rather on the exacerbation process triggered by the withdrawal of bDMARDs that induced deep remission for a prolonged period. It is intriguing to examine the differences in cytokine kinetics between TNF inhibitor users and IL-6 inhibitor users on the path to remission, as well as variations between those who achieve remission and those who do not. It is conceivable that significant differences may exist between TNF inhibitor users and IL-6 inhibitors users when this kind of study is conducted. Once deep remission was achieved, the exacerbation process may resemble the disease developing process from healthy state. This is one of the reasons why there are no specific patterns of cytokine kinetics in the process of exacerbation after a prolonged period of deep remission. This is merely a hypothesis, and as such, we do not include it in the paper.

---

## [Decision Letter · Decision Letter 1]

12 Feb 2024

Biomarker combination predicting imminent relapse after discontinuation of biological drugs in patients with rheumatoid arthritis in remission

PONE-D-23-22054R1

Dear Dr. Minota,

We’re pleased to inform you that your manuscript has been judged scientifically suitable for publication and will be formally accepted for publication once it meets all outstanding technical requirements.

Kind regards,

Masataka Kuwana, MD, PhD

Academic Editor

PLOS ONE

Additional Editor Comments (optional):

Reviewers' comments:

Reviewer's Responses to Questions

**Comments to the Author**

1. If the authors have adequately addressed your comments raised in a previous round of review and you feel that this manuscript is now acceptable for publication, you may indicate that here to bypass the “Comments to the Author” section, enter your conflict of interest statement in the “Confidential to Editor” section, and submit your "Accept" recommendation.

Reviewer #1: All comments have been addressed

Reviewer #2: All comments have been addressed

2. Is the manuscript technically sound, and do the data support the conclusions?

Reviewer #1: Yes

Reviewer #2: Yes

3. Has the statistical analysis been performed appropriately and rigorously? 

Reviewer #1: Yes

Reviewer #2: Yes

4. Have the authors made all data underlying the findings in their manuscript fully available?

Reviewer #1: Yes

Reviewer #2: Yes

5. Is the manuscript presented in an intelligible fashion and written in standard English?

Reviewer #1: Yes

Reviewer #2: Yes

6. Review Comments to the Author

Reviewer #1: I consider that the authors have revised the manuscript almost properly according to reviewer’s comments.

Reviewer #2: The authors have effectively addressed the concerns I highlighted in my review.

While there might be a selection bias in the target population (the number of patients who opted to either continue bDMARD treatment or discontinue without participating in the study was not recorded), the limitations section highlights the future requirement to verify reproducibility through a validation cohort.

This study, which predicts relapses after discontinuing bDMARDs using biomarkers and potentially facilitates the timely restart of biodegradable DMARDs, is intriguing.

7. PLOS authors have the option to publish the peer review history of their article (what does this mean?). If published, this will include your full peer review and any attached files.

Reviewer #1: No

Reviewer #2: **Yes: **Shinji Watanabe

---

## [Editor Report · Acceptance letter]

13 Mar 2024

PONE-D-23-22054R1 

PLOS ONE

Dear Dr. Minota, 

I'm pleased to inform you that your manuscript has been deemed suitable for publication in PLOS ONE. Congratulations! Your manuscript is now being handed over to our production team.

Kind regards, 

on behalf of

Prof. Masataka Kuwana 

Academic Editor

PLOS ONE